Context dependent prediction in DNA sequence using neural networks

http://orcid.org/0000-0001-7440-0718 Grønbæk Christian 1 2 christian.groenbaek@gmail.com
Liang Yuhu 3
Elliott Desmond 3
Krogh Anders 3 4
1 Department of Biology, University of Copenhagen , Copenhagen , Denmark
2 Novo Nordisk Foundation Center for Basic Metabolic Research, Faculty of Health and Medical Science, University of Copenhagen , Copenhagen , Denmark
3 Department of Computer Science, University of Copenhagen , Copenhagen , Denmark
4 Center for Health Data Science, University of Copenhagen , Copenhagen , Denmark
Shang Yuan
Electronic publication date: 2022 Sep 20
Publication date: 2022
Volume: 10
Electronic Location ID: e13666
Received 2022 Jan 4; Accepted 2022 Jun 10
Copyright: © 2022 Grønbæk et al.
Copyright year: 2022
Copyright holder: Grønbæk et al.
License: This is an open access article distributed under the terms of the Creative Commons Attribution License, which permits unrestricted use, distribution, reproduction and adaptation in any medium and for any purpose provided that it is properly attributed. For attribution, the original author(s), title, publication source (PeerJ) and either DOI or URL of the article must be cited.
License URL: https://creativecommons.org/licenses/by/4.0/

Keywords: DNA, Neural networks, Predictability, Patterns, Signals of periodicity

Funding: Novo Nordisk Foundation through the MLLS Center NNF20OC0062606 Novo Nordisk Foundation This work was supported by the Novo Nordisk Foundation through the MLLS Center (Basic Machine Learning Research in Life Science, NNF20OC0062606). The GPU servers were also supported by a grant from the Novo Nordisk Foundation. The funders had no role in study design, data collection and analysis, decision to publish, or preparation of the manuscript.

==============================
One way to better understand the structure in DNA is by learning to predict the sequence. Here, we trained a model to predict the missing base at any given position, given its left and right flanking contexts. Our best-performing model was a neural network that obtained an accuracy close to 54% on the human genome, which is 2% points better than modelling the data using a Markov model. In likelihood-ratio tests, the neural network performed significantly better than any of the alternative models by a large margin. We report on where the accuracy was obtained, first observing that the performance appeared to be uniform over the chromosomes. The models performed best in repetitive sequences, as expected, although their performance far from random in the more difficult coding sections, the proportions being ~70:40%. We further explored the sources of the accuracy, Fourier transforming the predictions revealed weak but clear periodic signals. In the human genome the characteristic periods hinted at connections to nucleosome positioning. We found similar periodic signals in GC/AT content in the human genome, which to the best of our knowledge have not been reported before. On other large genomes similarly high accuracy was found, while lower predictive accuracy was observed on smaller genomes. Only in the mouse genome did we see periodic signals in the same range as in the human genome, though weaker and of a different type. This indicates that the sources of these signals are other or more than nucleosome arrangement. Interestingly, applying a model trained on the mouse genome to the human genome resulted in a performance far below that of the human model, except in the difficult coding regions. Despite the clear outcomes of the likelihood-ratio tests, there is currently a limited superiority of the neural network methods over the Markov model. We expect, however, that there is great potential for better modelling DNA using different neural network architectures.

Introduction

We considered the question of how predictable chromosomal DNA is from context. That is: how well can one predict the base at any given position in the DNA, if the surrounding context of bases around the position are known, e.g., the five bases to the left and right of the position. The motivation for considering this question is that the probability of observing a base in a certain position in the human genome is highly context-dependent. It is thought that through evolution the genomic DNA has been shaped by mutational processes. The cellular machinery for error correction and repair has left its marks in the nucleotide composition along with structural and functional constraints.

In our parallel work (Liang et al., 2022), we use conditional probability models with contexts up to 14 nucleotides on each side around the central nucleotide (29 nucleotides in total). It is likely that larger contexts influence the probability of a nucleotide, since many structures in the genome, such as genes, nucleosomes, and transposable elements, are larger than 29 nucleotides. With these conditional probability models we are forced to use contexts of smaller size as input, because the number of parameters otherwise explodes. In the present study we used neural networks to model nucleotide contexts, which allowed for larger contexts. We then compared the performance of the neural network-based models to that of our simpler conditional probability models.

Concretely, we formulated the question in terms of estimating the probability of the four possible bases at a given position in the genome, conditioned on the knowledge of the bases in flanks of equal length to the left and to the right of the position (Liang et al., 2022). Using common notation these probabilities can be written

p(basei=x|contexti)

where basei is one of the four bases, A, C, G and T, and the contexti is given by the left flank and right flank, which are the sequences of bases to the left and to the right of position i. Having modelled these probabilities, we let the model predict the missing base as the most probable assignment basei∗:

basei∗=argmaxx∈A,C,G,Tp(basei=x|contexti)

To measure the performance of the models we used the accuracy, which is the fraction of all positions at which the model’s prediction is correct, i.e., where the model assigns the highest probability to the true base. The neural network models were trained to minimize a categorical cross-entropy loss, which is standard in classification tasks.

Our baseline model (Liang et al., 2022), referred to as the k = 3 central model, used contexts of size 6, i.e., three bases in the left and right flanks around the missing position. The model parameters were estimated simply by counting: for a given context, we found the number of times each of the four letters occured in the central position throughout the genome; the corresponding frequencies could then be taken to be the desired conditional probabilities (this gives 3 × 46 ∼ 12,000 parameters). On the human genome, the k = 3 central model obtained an accuracy of ∼38%. We then observed that increasing the flank size led to improved performance. With a Markov model estimated on the left-hand flanks of size k = 14 the accuracy reached ∼52%, while for the k = 3, 4, 5 central models the level was below 50%. In terms of number of parameters these models were not economic; with the k = 5 central model the number rose to ∼3 million, and with the Markov k = 14 model it rose to ∼0.8 billion.

Given this behaviour, it was natural to consider whether improved accuracy could be achieved using bigger contexts, e.g., 50 or 100 bases. However, with the k central and Markov models, the frequencies used to estimate the model’s parameters over such large contexts resulted in statistics that were too sparse: The model parameters outnumbered the input data points. We turned to neural networks language models to overcome this curse of dimensionality (Bengio et al., 2003). We constructed our neural networks so they could process larger contexts than the previous models and yield a single prediction for the value of the missing base. For defining the larger architecture of the neural networks relevant to our task, an obvious source of inspiration was their use in natural language modeling tasks. In that field, recurrent neural networks with long short term memory hidden units (LSTM) and Transformer networks have been successful (Hochreiter & Schmidhuber, 1997; Vaswani et al., 2017). We also carried out experiments using feed forward networks and convolutional networks (Fukushima, 1980; Lecun et al., 1998).

Within genomics these types of deep networks have all been used, alone or in combination, and many different tasks have been approached. Several applications fall within functional genomics, but the field covers much more. We refer the reader to the perspective by Zou et al. (2018) and references therein. Regarding the Transformer, the ‘DNABERT’ model (Ji et al., 2021) can presumably be applied to the task we have considered, and it would be interesting to know its performance (our work with the Transformer was limited, mostly because of the delicacy in its training and hardware demands).

The best performing networks we found consisted of one or two convolutional layers for “word encoding” (tri- or quadro-nucleotides) followed by bi-directional LSTM layers. On the human genome, the accuracy of the best-performing model went beyond 53%. An additional ∼0.5% could be obtained by applying the model simultaneously on both strands and averaging (a bi-directional average which we also use with the Markov model). To formally compare the best neural network with the simpler models we conducted a likelihood-ratio test for non-nested models (Vuong, 1989).

We also trained the same or similar models on genomes of other organisms, which resulted in highly varying accuracy. In all cases the level was far above that of random guessing, which lay within 25–27% for the species we considered (taking base composition into account, Supplemental Tables and Plots, Table SI12). One option here was to obtain a measure of genomic similarity by applying a model trained on one organism’s genome to another organism’s genome. As an example we compared (a) the predictions obtained by applying a model trained on mouse DNA to human DNA to (b) the predictions obtained by applying a model trained on human DNA to the same DNA. The results were somewhat surprising: of the annotated matter we considered, the two models seemed to be in agreement only in coding regions, while elsewhere there were large differences, in particular in repeats.

It is also of interest to look into where a model performs well. For instance, it is to be expected that much of the accuracy obtained is due to repetitive DNA sequences. We show how the accuracy varies over annotated matter, such as various types of repetitive sequences, coding regions and more. Using Fourier transformation, we also investigated whether the predictions of a model carried signs of periodic components (not to be confused with periodicities in the sequence itself). This depended on the considered organism: in the genomes considered here we found clear signals only in human, persistently shared across the chromosomes. On the scale of the highest amplitudes in the Fourier coefficients these signals were small. For the mouse genome such signals surprisingly much less clear, verging on non-existent; in the zebrafish, yeast and fruit fly genomes we saw no consistent signs of this kind.

In the human genome we found quite similar signals in GC/AT content (which to the best of our knowledge have not been reported before; again these are not to be confused with periodicities in the sequence itself). The GC/AT period ranges are slightly different and the amplitudes also differ from those we found in the models’ predictions. The latter periods quite clearly could suggest connection with DNA wrapping/condensation (nucleosome distance and length of linker DNA), underpinned by the similarity to the GC/AT content case. In the other genomes we found some vaguely similar structure in the GC/AT content only in mouse, pointing in other directions than nucleosome arrangement. We did however find rather clear signs of a ∼10 bp periodic signal in all the species we considered.

Methods and implementation

We include here only a short description of the method used; for a more detailed version please see the dedicated Supplemental Methods.

The neural networks all take a short sequence of DNA as input, consisting of the context around a given position in the genome (DNA sequence of e.g., human). We used the common one-hot encoding to transform the letter sequence into a quantitative representation. The output of the network is a four dimensional real vector that followed the same encoding and for which the entries were positive and sum to 1. This property of the output was the result of letting the final layer of the network have a “soft-max” activation function. The output could then be interpreted as a probability distribution over the four letters; so for instance the first entry was the probability of having the letter A given the input. Activation functions were elsewhere set to the “rectified linear”, sometimes referred to as ReLu.

The networks were trained on a randomly sampled sizeable fraction of the given genome (single strand), about one third in the cases of human, mouse and zebrafish. The training process was divided into a series of “rounds”, each finished by a validation at that stage. The learning objective was to minimize a categorical cross-entropy loss (standard for labeling tasks). The final trained models were applied to the full genomes (i.e., tested on all positions). Two models were though trained and tested on truly split data.

We implemented the neural networks using Keras/Tensorflow (Tensorflow 2018, https://www.tensorflow.org), an open-source machine learning platform in Python. Among many great facilities, this includes automatic off-loading of core computations to Graphical Processing Units (GPUs). Our server ran with 12 GB RAM dedicated to the GPU, an Nvidia Tesla P100.

The Python code used for the paper is available in the GitHub, https://github.com, repository: ceegeeCode/randomodnar/. A notebook containing a walk-through of applying the code to the case of yeast is included.

Results

Human genome

We trained three multilayer neural network models of convolutional-LSTM type (LSTM200, LSTM200early, LSTM50) on the human reference genome hg38, and compared their accuracy when applied to the same genome. An additional model, LSTM50S, was trained only on the odd-numbered chromosomes of hg38. The LSTM200 model used flanks of 200 bases; LSTM200early was identical to LSTM200 but trained less; LSTM50 had almost the same architecture and LSTM50S was identical to LSTM200, but both used flanks of 50 bases. The reason for including LSTM200early was to investigate where the increase in accuracy from the extra training stemed from, while LSTM50 and LSTM50S were included to shed light on the impact of the context size. However, the main role of LSTM50S was to investigate if the inclusion of the training data in the test data resulted in exaggerated performance (we typically applied the models to the full genomes). With the same purpose we also trained and tested a model identical to LSTM50, but using a different split of the human genome to obtain non-overlapping training and test data (LSTM50P). The results showed that the accuracies we report on the other models (LSTM200, LSTM50 etc.) are hardly inflated, and if so by no more than ∼0.1% point (see Supplemental Methods, Section “On the models’ ability to generalise on unseen data”).

Accuracy

Figure 1 shows the performance of the models in terms of accuracy per autosomal chromosome (all values can be found in Table SI1 of Supplemental Tables and Plots). The LSTM200 showed invariably the best performance (overall ∼53%), but was followed closely by LSTM50 and LSTM50S; last was LSTM200early. The very same pattern was seen when the accuracy was grouped by annotation (Fig. 2).

Figure 1 Model accuracy comparison across the human autosomal chromosomes, hg38.

The named models are described in the text. The dashed line indicates the average performance of the LSTM200. All values can be found in Table SI1, Supplemental Tables and Plots.

Figure 2 Five model accuracy comparison by annotation in hg38.

All values can be found in Table SI2 in the Supplemental Tables and Plots.

The differences between LSTM200 and LSTM50 suggest that there is additional information in the larger contexts. Except for a more rich “word-encoding” convolutional part, LSTM200 differed from LSTM50 only in the use of larger flanks. The models had almost the same number of trainable parameters (LSTM200 2,227,582 vs. LSTM50 2,135,806). The benefit of the larger flanks as also supported by the results of LSTM50S, which were only marginally different from those of LSTM50. As expected LSTM50S performed a little better than LSTM50 on the odd-numbered chromosomes, while on the even-numbered chromosomes LSTM50S did a little worse. The level of the differences was, however small, ∼0.1–0.2% points (see Supplemental Tables and Plots for results of LSTM50S and Supplemental Methods, Table SA1 and Fig. SA4–6, for comparisons to the other models).

The differences in performance between LSTM200 and LSTM50S stemed from the different flank sizes and the differences in their training sets. We also saw in experiments that shuffling the contexts while retaining some few central bases led to some loss in accuracy (data not included).

The differences between LSTM200 and LSTM200early showed broadly that the increase in performance from LSTM200early to the (much) more trained LSTM200 was not ascribable to particular chromosomes or annotated parts of the genome: The increase was about the same across chromosomes and across annotations.

Figure 2 shows that, as expected, a large part of the models’ accuracy was obtained in the repeats, where all models invariably performed much better than elsewhere. Whereas the total accuracy (‘all’) was slightly above 50%, it was about 65% in the repetitive sequences as defined by the repeat masking in the downloaded genome sequence (UCSC 2018, https://hgdownload.cse.ucsc.edu). The difficulties in predicting were most pronounced in coding regions, though the accuracy was still close to 40% there. All values can be found in Supplemental Tables and Plots, Table SI2.

As can be seen in Fig. 3, the accuracy level appeared stable across the chromosomes (only data for LSTM200 are included). The largest variation was seen for simple repeats. There were some differences between (the GC rich) chromosome 19 and the other chromosomes across the annotations. Also notable, the ‘gene’ category appeared very close to the total, ‘all’. All values can be found in Supplemental Tables and Plots, Table SI3.

Figure 3 LSTM200 accuracy by annotation per chromosome, hg38.

Cross-species prediction

Here we used the models to study the similarity of the human and mouse genomes: if the two genomes are highly similar, we expect the performance of a model trained on the mouse genome, when applied to predicting on the human genome, to be very close to that of a human-model. In a general perspective, a measure of genome similarity could be obtained in this way, e.g., by a likelihood-ratio test of the models or similarly.

For the concrete task, we trained a model with architecture identical to that of LSTM50 on the mouse reference genome, mm10. We refer to this (trained) model as mouseLSTM50. We return below to describe its performance on the “host” genome (GRCm38/mm10); in this section we consider its performance when applied to the human reference genome, hg38.

As Figs. 1 and 2 clearly show, mouseLSTM50 did not perform nearly as well as LSTM50 on hg38. The overall accuracy of mouseLSTM50 lay at about 43%, and varied between about 42% and 48% over the chromosomes (Table SI6, Supplemental Tables and Plots). Remarkably only in the coding parts did mouseLSTM50 perform at par with LSTM50. In other annotated matter the accuracy of mouseLSTM50 was lower, the biggest difference being in repeat sequence where the overall accuracy was at a mere 46.9%, far from the 63.7% of LSTM50. These values are further supported in Fig. 4, which covers four annotations. What is shown is scatter plots of the probability of the reference-base assigned by the two models, summarized as densities in a grid obtained by binning the two axes. Here two models agreeing would show up as a clear diagonal, while disagreements are seen off the diagonal. The brighter an off-diagonal patch, the more substantial the difference (a pronounced case is seen for repeats, the plot furthest to the right). A bright diagonal and only little disagreement is seen only for the coding sections. Similar plots for chromosomes 17, 18 and 19 are placed in the Supplemental Tables and Plots, Fig. SI1.

Figure 4 LSTM50 vs. mouseLSTM50 on human.

Density-plot of probabilities of reference bases in four annotated parts of chromosome 1 of the human reference genome, hg38, according to LSTM50 (x-axis) and mouseLSTM50 (y-axis). The density was obtained in a grid defined by 50 equally sized bins on each axis. From upper left, (A) all positions (‘all’), (B) ‘gene’, (C) coding sequence (‘cds’), (D) repeat masked (‘Rmsk and trf repeats’). Please note that colors are not shared between the plots.

Signals of periodicity in predictions

In addition to examining the models’ performance with respect to the discrete annotations, we looked into whether prediction as a function of genomic position showed signs of periodicity.

In our analysis we segmented each chromosome into adjacent 1 Mb segments and considered at each position the probability assigned by the model to the reference base (the true base). The 1 Mb size was chosen for practical reasons (we wanted a local genomic scale and at the same time to limit the number of plots). Segments with more than 10% disqualified positions (contexts containing at least one N) and short parts of less than 1 Mb were left out (leaving typically more than 95% of the segments included; the number of segments for each chromosome and other relevant statistics can be found in Supplemental Data and Data Checks). To look for periodic signals we Fourier transformed each of these 1 Mb long arrays of positive numbers (between 0 and 1). In general, no frequencies were seen to carry particular weight. However, when we took the norm of all Fourier coefficients within a sliding window of a set amount of frequencies (typically 1,000 and a step size of 100) clear patterns emerged as those shown in Fig. 5 (similar plots for the other chromosomes can be found in Supplemental Human Fouriers). To check for the possibility of computational artifacts we shuffled the input arrays and saw no such patterns in the output. The same happened when we randomized all positions, while the patterns resisted randomization in half of all positions or in the subset of disqualified positions (Supplemental Methods, Figs. SA9 and SA10). The same patterns found in the adjacent 1 Mb segments also appeared when Fourier transforming randomly picked segments of 1 Mb and other lengths and using other window sizes (data not included).

Figure 5 Fourier spectra on the reference-base probability array of LSTM1 for chromosome 20, hg38.

The chromosome is divided in 64 adjacent segments of 1 Mb; segment numbers along with quality fraction are indicated in the legend. Upper left plot (A) shows the results for all segments in this chromosome; the following seven plots (B–H) cover the segments in increasing order. The centromeric region lies in segments 26–30. Similar plots for all somatic chromosomes are placed in the Supplemental Fouriers Human.

First, as seen in Fig. 5, there were peaks at frequencies slightly above 4,000, 7,500, 11,000 and 18,000 (and maybe 14,000 too). With 1 Mb length these translate into windows of periods of ∼ (200 bp, 250 bp), (118 bp, 133 bp), (83 bp, 91 bp) and (52 bp, 56 bp) (and (67 bp, 71 bp)), respectively. If we assign a peak to the mid frequency in a window, the three peaks are at ∼ 222 bp, 125 bp, 87 bp and 54 bp (and 69 bp), respectively. These numbers, of which the first and the second are close to the sum of the two following them, appear to be quite close to the lengths of DNA sequences related to nucleosome wrapping: 147 bp wound on each nucleosome and with linker DNA of varying length ∼20–80 bp (see Baldi, Korber & Becker (2020) for a recent review).

Second, in all the chromosomes there were segments showing much more abrupt variation. These segments match very well the centromeric regions; in e.g., chromosome 20 (Fig. 5) these segments have numbers 27 through 29, which corresponds well with the centromeric region; for more examples see Supplemental Fouriers Human.

These periodicity signals were found not only with the best performing convolutional LSTMs, but also with the k = 5 central model and the Markov model (k = 14) of Liang et al. (2022), albeit with slightly larger span of amplitudes (see Supplemental Fouriers Human, simple models).

In higher frequency ranges we also found signs of periodicity (we only considered LSTM200 here). Thus, in the range 40,000 to 140,000 (see Supplemental Fouriers Human) a set of three or four peaks occur in all chromosomes, the most conspicuous peak at around 125,000 corresponding to a period of about 8 bp.

We wondered whether the signals could have some origin in GC/AT content, i.e., if there are similar signals in GC/AT composition. To that end we applied the Fourier transform to “boolean arrays indicating Gs and Cs in the DNA”: every 1 Mb DNA segment was encoded by leaving a 1 at all G/C-positions and a 0 at all A/T’s. The arrays were segmented and quality filtered as described above (same segmentation). Figure 6 shows an example of the output, clearly suggesting a connection to the signals in the reference-base probability arrays. Peaks were found at similar but somewhat shifted frequencies, and with the weight of the peaks being different from those seen in the models’ probability arrays (plots of Fouriers on GC/AT content for the other chromosomes can be found in the Supplemental Fouriers Human). Also in the high frequency range 40,000 to 140,000 this appeared to be the case (see Supplemental Fouriers Human); notably the most conspicuous peak in that range was found at slightly less than 100,000 corresponding to a period of a little more than 10 bp, close to the ∼10.4 period of the double helix.

Figure 6 Fourier spectra on the GC-content array for chromosome 20, hg38.

See caption in Fig. 5 for content.

It is also of interest to see if periodic signals occur in the reference-base probability arrays (on hg38) of the model trained on the mouse genome, mouseLSTM50. As seen in Fig. 7 this was the case, but with a variation that appeared less smooth than in the results from the human model (Fig. 5). As there we see five peaks, but here the first two appear at lower frequencies, while the following three were placed at similar frequencies. All five peaks had less amplitude, and distributed quite differently than in the human model case.

Figure 7 Fourier spectra on the reference-base probability array of mouseLSTM50 for chromosome chr20, hg38.

See caption in Fig. 5 for content.

Comparison to simpler models

To compare our best neural network model (LSTM200) with the simpler models from our parallel study (Liang et al., 2022), we worked out likelihood-ratio tests (Vuong, 1989), all for the null hypothesis of equally performing models. The tests were first done per chromosome by computing the test value on a random sample of 10% of the positions in the genomic sequence; a plot of the obtained test values can be found in the Supplemental Tables and Plots, Fig. SI2. A test value for the complete genome was then computed by aggregating the results for the chromosomes. We carried this out for the k = 3, 4 and 5 central models and for the Markov model (k = 14). In each case the simpler model acted as base (“denominator”) and the LSTM200 model as the “numerator” in the ratio test. The results shown in Table SI7, Supplemental Tables and Plots, reveal invariably the superiority of the LSTM200, and with test values increasing with the simplicity of the base model (with p-values essentially all zero a correction for multiple tests is unnecessary). Also, the test values per chromosome increased with the size of the chromosome (and, equivalently, to the sample size) as is evident from Supplemental Tables and Plots, Fig. SI2, as should be the case on a rejection of the null hypothesis (Vuong, 1989).

To illustrate these figures we made model-model plots as the one shown in Fig. 4, but here for each of the simple models vs. LSTM200 and covering the whole genome rather than particular annotations (Supplemental Tables and Plots, Fig. SI3). Recalling that brighter off-diagonal areas signify differences in performance, increasing with the brightness, these plots indicate that LSTM200 generally has higher confidence in its predictions than the simpler models. Thus, for the Markov model, there seems to be a “bend” of the brighter area in favor of LSTM200.

With an overall performance of the Markov model (k = 14) of about 52% (Liang et al., 2022) it is of further interest to consider how well the model handles the annotated categories as compared to the LSTM200. In Fig. 8 we show the Markov model-LSTM200 plots for some major annotations in hg38, chromosome 1 (similars plots for chromosome 17, 18 and 19 can be found in the Supplemental Tables and Plots, Fig. SI4). The same “bend” of the bright areas is again seen; for the coding sections the bright area even appears shifted above the diagonal. This indicates that the LSTM200 rather unconditionally is more confident than the Markov model in the coding regions.

Figure 8 Markov model (k = 14) (x-axis) vs. LSTM200 (y-axis) on four types of annotated matter in chromosome 1, hg38.

Each plot shows the density of the reference-base probabilities according to the named models within the given annotated matter. The density is obtained in a grid defined by 50 equally sized bins on each axis. From upper left, (A) all positions (all), (B) gene, (C) coding sequence (cds), (D) repeat masked (Rmsk and trf repeats). Please note that colors are not shared between the plots.

In our parallel work (Liang et al., 2022) we also applied our models to sets of genetic variants and built a model of mutability. We carried out a subset of the same analyses here, but based on the LSTM200. The results (Supplemental VII) match those of our parallel work.

Other genomes

We trained and applied models identical to the LSTM50 on reference genomes of yeast (S.cerevisiae), fruit fly (D.melanogaster), mouse (M.musculus) and zebrafish (D.rerio).

Accuracy

As reported in Table 1 the obtained accuracy varies much across these organisms. For yeast and fruit fly we report here the results on the early stage models, LSTM50early (Supplemental Methods). For yeast the genome size appears to be too small for sufficient training of a model with about 2 million parameters. We observed that training the model more extensively resulted in overfitting, which increased with the amount of training (Supplemental Tables and Plots, Fig. SI5, Supplemental Methods Fig. SA8). An identical model trained on a truly split genome achieved in validations an accuracy level of about 40%, a better result, but still low (LSTM50P for yeast, Supplemental Methods Fig. SA8). Similarly, in the case of the fruit fly we observed overfitting in the more extended training, albeit much less than with yeast (Supplemental Tables and Plots, Fig. SI6). The result for fruit fly in Table 1 is therefore likely to the low side. For mouse and zebrafish the performance was quite in line with that for the human genome (LSTM50); the higher level for zebrafish is apparently partially stems from repeat sequence (repeats consume about 50% of the zebrafish genome).

Table 1 Overall accuracy for the LSTM50 trained on reference genomes of the named organisms. The numbers for yeast and fruit fly are due to the early stage models, LSTM50early.

Chr/annotation	All	RepeatsGenomeSeq	
yeast (S.cerevisae)	0.3718	0.4476	
fruit fly (D.melanogaster)	0.4188	na	
mouse (M.musculus)	0.5081	0.6644	
zebrafish (D.rerio)	0.5428	0.7024	

While we did not consider the more narrowly annotated matter, we looked into whether or not periodic signals were found in the predictions also for these genomes.

Signals of periodicity: in mouse? in zebrafish?

Broadly, for the yeast, fruit fly and zebrafish genomes we found no persistent signals across the chromosomes, either in the reference-base probabilities or GC/AT content. One exception was though a rather clear peak (less in yeast than in fruit fly and zebrafish) corresponding to a period of about 10 bp, i.e., close to the turn of a DNA helix. Most notable apart from this is maybe that in fruit fly and yeast the decay of the coefficients varies substantially across the segments (between two extremes, roughly). More surprisingly, also in the case of mouse we found no clear persistent signals as those in the lower frequency range in the human case. However, there was some “bundling” at particular frequencies: for reference-base probabilities at frequencies around 5,000 (but very weakly) and for GC/AT content (more strongly, but not strong) around 7,500 and, maybe, around 12,500, the segments appear more tightly bundled than elsewhere. Figure 9 shows the plots resulting from the very same Fourier analysis that we carried out for the human case, but for chromosome 11 of the mouse reference genome, mm10 (results for other chromosomes can be found in the Supplemental Fouriers mouse). In the higher frequency range (40,000 to 140,000) there seems also to be some structure in GC/AT content, the same three peaky regions reappear across the chromosomes as in the human genome (at around 60,000, 100,000 and 120,000).

Figure 9 mouseLSTM50 and GC/AT content on mouse genome (mm10).

Fourier spectra on reference-base probability and GC/AT content for chromosome chr11. The chromosome is divided in adjacent segments of 1 Mb. Left-hand plot (A) shows the results for the mouseLSTM50 predictions in all segments in this chromosome, the right-hand plot (B) shows the similar results for the GC/AT content.

Discussion

Our motivation for using neural networks for the task of “predicting DNA from context” was that larger flanks than those within reach by simple models as in Aggarwala & Voight (2016) and Liang et al. (2022) could contain information that would allow improvement of the prediction. For each k = 3, 4, 5 the central model (Liang et al., 2022) provided an upper bound for the overall accuracy using flanks of size k and our neural networks outperformed these clearly. Also, LSTM200, which used flanks of size 200, outperformed the flanks-50 LSTM50 (these models were almost identical). So our main hypothesis seems right. To further support this we earlier ran tests shuffling and randomizing parts of the contexts, and saw the expected decrease in performance (data not included). However, while our models broke the 50% mark, it was seemingly not by much, and our very best neural net added only about 2% points to the accuracy of the Markov model (k = 14). We have though noted that obtaining these extra few percentage points appeared to be a struggle; about two thirds of the complete training of LSTM200 were spent there. These extra percentage points were obtained across all structural parts (annotations), and not only by an improvement in e.g., repetitive sequences (compare LSTM200early and LSTM200 in Fig. 2). Further, the likelihood-ratio tests showed that the LSTMs are superior to the simpler models, with clear significance. This is a strong sign that the LSTMs are hitting the conditional probability distribution of the bases closer, and indeed are optimizing the categorical cross-entropy and not the accuracy. In terms of number of parameters, one facet of model complexity, the LSTM200/50 was the lightest with about 2.3 million; the k = 5 central model had about 3.1 million parameters and the Markov k = 14 model about 0.8 billion, i.e., about 350 times more than our neural networks. So if the aim is not to improve model performance and the number of parameters is not an issue, the Markov model comes out as the model of choice, due to its simplicity, good performance and ease of use. As opposed to the neural networks, the performance of the Markov model is however “saturated”, because it is estimated on the full genome. Our experiments do not show that our “hi-score” (about 54%) is a bound that neural nets cannot push; to the contrary it would be surprising if that would not be doable.

Considering the level of the accuracy obtained on the human genome, it does catch the eye that it ends up near 50%, and we have noticed that this level is close to the accuracy reached on the “gene” and “intron” annotated part. This probably reflects that introns dominate these regions and their composition is very similar to intergenic regions. Looking into how the accuracy was obtained, we expected that the prediction would be easier in repetitive sequences than elsewhere. This was clearly the case, but in all (other) annotated parts the performance was far better than by random guessing (Fig. 2). Variation across chromosomes (Fig. 1) and chromosomes/annotations were rather small (Fig. 3). Comparing the accuracy on the odd-numbered vs. the even-numbered chromosomes though revealed a difference shared by the models (Supplemental Methods, Table SA2 and Fig. SA4).

When training and running models on other genomes we saw performance at the same level as that for the human genome (mouse, zebrafish), while achieving much lower accuracy for smaller genomes (fruit fly, yeast). We did not try to explore the sources of the performance for these genomes as far as we did for the human genome. Repetitive sequences though played a similar role. For all these cases we used the architecture of the LSTM50; thus the trained models were identical except for the values of the trainable parameters. This allowed direct comparison, but to avoid overfitting the amount of training had to be kept much lower for the small genomes (yeast, fruit fly) than for the larger genomes (human, mouse, zebrafish), which could all be trained to similar extent. The modest performance in the cases of yeast and fruit fly appears to be due to too few data points as compared to the number of model parameters. This calls for experiments using more economic models.

Having models trained on different organisms opens a possibility for comparison of their genomes. We found it particularly interesting to apply the model trained on the mouse genome (mouse-model, mouseLSTM50) to the human genome and then compare to the human-model (LSTM50). This gave the rather surprising result that apart from the coding regions, the mouse-model was performing substantially worse than for the identical model trained on the human genome (LSTM50). The congruence on the coding part hangs well together with the known genetic similarities between mouse and human. The differences can have many sources. For one, the repetitive sequences in the two genomes may be very different. Certainly, as the models fall long short of predicting their host DNA perfectly, their “representation” of that DNA may have large imperfections, and possibly specific to the DNA in question. At the extreme, comparing two essentially random models would be illogical; however, our models are clearly very far from such a state.

That this is indeed the case is well supported by the obtained accuracy and the likelihood-ratio tests that we have carried out, but also by the periodic signals that we have found. In the case of the human genome, but hardly in mouse, the periodicities in the reference-base probabilities seem connected to those seen in the GC/AT content. In the human case the periods in these are very similar and in the mouse the same kind of “bundling” appeared in both (though at somewhat different frequency ranges, and only very weakly in the reference-base probabilities). This revealed that, at least to some extent, the model is capable of capturing this structure. The differences between the GC/AT content and the models’ predictions could have several sources: the model has not picked up perfectly the GC/AT composition bias; there may exist other periodic patterns to which the model has adapted.

Regarding the method, applying Fourier transformation in various ways to DNA sequences is certainly not new. Thus e.g., in Voss (1992) and maybe earlier, Fourier transform was applied to DNA sequence. As mentioned we are not seeking periodicities in the sequence itself, but Fourier transform quantitative arrays of model probabilities and GC/AT indication; our approach to understanding the output can be summarized as considering “the cumulative power spectrum in a running window”. A relatively recent application of the Fourier transform to DNA and aimed at genome comparison, consists in considering the cumulative power spectrum of each of the indicator functions of the four bases (Dong et al., 2018; Pei et al., 2019). In between Voss (1992) and these publications much has though been done in spectral analysis of DNA sequences by Fourier transforms, see e.g., Lobzin & Chechetkin (2000), Widom (1996) and references therein. The well-known ∼10.4 bases periodicity in dinucleotide composition which has been linked to histone binding, has also called for Fourier analyses but aimed at di- or trinucleotide composition (Trifonov & Sussman, 1980; Widom, 1996; Yuan & Liu, 2008). The work of Lobzin & Chechetkin (2000) includes also a rigorous statistical analysis for significance in spectra, something we have not touched upon. Our findings are obtained by visual inspection. As mentioned, analyses similar to those of GC/AT content that we have carried out and reported here we have not come across in the literature. To rule out the possibility of mere computational artifacts, we did simple shuffling and randomization tests. Also, the periodicity signals vary substantially over the organisms considered. The patterns seen in GC/AT content rule out a modeling artifact, as does the fact that the reference-base probabilities of the Markov model show almost identical signs of periodicity as for the LSTM200 on human (chromosome 22). Finally, in human the characteristic patterns over the segments were seen to change abruptly in centromeric regions; in mouse this phenomenon appeared absent well in line with the fact that centromeres in mouse are found at the ends of the chromosomes with essentially no short chromosome arm.

We aim to dedicate a paper to exploring a possible biological explanation for these patterns, but some treatment is called for here. The periods that we have found in the case of the human reference genome could point to the chromatin’s “beads-on-a-string” structure, in which the DNA is wound up over nucleosomes at quite regular spacing. This structure is though probably local, dynamic, and may vary much across a chromosome. Notwithstanding, we found clear signs of periodic regularity, with periods shared by all autosomal chromosomes. The similarity to the periodic signs in GC/AT content could support a connection to nucleosome arrangement (Hughes & Rando, 2009), as do the abrupt changes in the patterns in segments covering centromeres (human) (Sullivan & Karpen, 2004); see also the more recent review by Chereji & Clark (2018).

On the other hand, patterns with these periods showed up only in the human genome. In mouse there was a certain “bundling” in the GC/AT content case, but only very vaguely in the reference-base probabilities. For the other organisms’ genomes that we considered, similar signals were essentially absent, except for a common, though not equally clear, peaky region corresponding to periods of about 10 bp. These hint at the well-known ∼10.4 bases periodicity in dinucleotide occurence, accepted to be connected to nucleosome positioning. However, the clear, longer periods only appear in human, which seems to undermine our nucleosome-positioning hypothesis, since nucleosome organization is regarded to be quite similar across species. These periodicities are hardly multiples of the ∼10.4 bases repeat pattern: were this the case, a series of corresponding frequencies should be expected, which appears quite clearly not to be the case. Also, in the other genomes we should then have observed such “harmonic series”. So possibly the signals in human have other sources or an in-between—mechanisms of nucleosome positioning may differ across species.

As a final remark, we can, strictly speaking, only say that we have seen these signals in a reference genome. A complete lack of a biological source seems highly unrealistic. At the other extreme, if the periodic patterns in the human genome are connected to nucleosome positioning, it could be a piece in the puzzle of the gene regulation in non-coding material in humans (and possibly for other species). More broadly, since these patterns appear to be different between species, they could very well be of benefit in several areas.

Another direction to pursue is that of making use of the representation of DNA that our models inherently learn. To this end one simply taps the output from one of the hidden layers of the model, e.g., the final dense layer. We only experimented very little with making use of these inherent word-representations. An existing work (Ng, 2017) was brought to our attention, in which one of the word-encodings of word2vec (Mikolov et al., 2013) is applied to DNA k-mers. Elegantly, the obtained word-representation is then used for a very fast alternative method for alignment of DNA-sequences, simply by taking dot-product of word-vectors. The same approach could be used with our models, but we suspect that they retain far more information about a few bases around the central position of each context. Probably the inherent representations will not result in quite so favourable comparisons to the Needleman-Wunsch alignment method as found in Ng (2017).

Several referees rightly pointed out that it is unfortunate that the training and test sets overlapped. When we started this project, we considered the three billion positions in the human genome as an infinite pool of samples. As training times grew, this approximation became less convincing. Since we did not have the resources to rerun everything, we tested with one of the models (LSTM50P) and found no discernible overfitting. From comparison of the models we also concluded that there were hardly any signs of inflation due to overlapping contexts in test and training sets. A cautious estimate of the exaggeration of the models’ accuracy ended at ∼0.1% point. We therefore believe that the stated accuracy figures represent fair estimates of the models’ predictive power.

Data

For all training, testing and prediction we used reference genomes downloaded from major resources (NCBI 2018, https://www.ncbi.nlm.nih.gov; UCSC 2018, https://hgdownload.cse.ucsc.edu). For more details on the downloads along with some statistics and checks of the data see Supplemental Data and Data Checks.

Supplemental Information

Supplemental Information 1 Supplementary Methods.

Click here for additional data file.

Supplemental Information 2 Data and data checks.

Click here for additional data file.

Supplemental Information 3 Supplementary tables and plots.

Click here for additional data file.

Supplemental Information 4 Fouriers, human genome, LSTM1, low frequencies.

Click here for additional data file.

Supplemental Information 5 Part 1: Fouriers, human genome, LSTM1, higher frequencies.

Click here for additional data file.

Supplemental Information 6 Part 2: Fouriers, human genome, LSTM1, higher frequencies.

Click here for additional data file.

Supplemental Information 7 Fouriers, human genome, GC/AT content, low frequencies.

Click here for additional data file.

Supplemental Information 8 Part 1: Fouriers, human genome, GC/AT content, higher frequencies.

Click here for additional data file.

Supplemental Information 9 Part 2: Fouriers, human genome, GC/AT content, higher frequencies.

Click here for additional data file.

Supplemental Information 10 Fouriers, human genome, simple model; mouseLSTM4.

Click here for additional data file.

Supplemental Information 11 Fouriers, mouse genome, LSTM1, low frequencies.

Click here for additional data file.

Supplemental Information 12 Fouriers, mouse genome, LSTM1, higher frequencies.

Click here for additional data file.

Supplemental Information 13 Fouriers, mouse genome, GC/AT content, low frequencies.

Click here for additional data file.

Supplemental Information 14 Fouriers, mouse genome, GC/AT content, higher frequencies.

Click here for additional data file.

Supplemental Information 15 Fouriers, zebrafish genome, LSTM1, low frequencies.

Click here for additional data file.

Supplemental Information 16 Fouriers, zebrafish genome, LSTM1, higher frequencies.

Click here for additional data file.

Supplemental Information 17 Fouriers, zebrafish genome, GC/AT content, low frequencies.

Click here for additional data file.

Supplemental Information 18 Fouriers, zebrafish genome, GC/AT content, higher frequencies.

Click here for additional data file.

Supplemental Information 19 Fouriers, fruit fly genome, LSTM4.

Click here for additional data file.

Supplemental Information 20 Fouriers, fruit fly genome, GC/AT content.

Click here for additional data file.

Supplemental Information 21 Fouriers, yeast genome, LSTM4.

Click here for additional data file.

Supplemental Information 22 Fouriers, yeast genome, GC/AT content.

Click here for additional data file.

Supplemental Information 23 SNP analysis.

Click here for additional data file.

This work could not have been done without access over a considerable period of time to servers with ample storage space and fine GPU facilities. It is therefore a pleasure to have the opportunity to thank Robin Andersson and Albin Sandelin at the Department of Biology of the University of Copenhagen for generously allowing us to run our analyses on their GPUs virtually ad libitum and to Hanne Munkholm for server hospitality and support to make it possible in the first place. In the same vein, Christian Grønbæk wants to thank Nicolas Alcaraz for help with the GPU set up early on. It is also a pleasure for us to thank Piero Fariselli for initial discussions and for sharing his material with us.

Additional Information and Declarations

Competing Interests

Author Contributions

Data Availability

The authors declare that they have no competing interests.

Christian Grønbæk conceived and designed the experiments, performed the experiments, analyzed the data, prepared figures and/or tables, authored or reviewed drafts of the article, and approved the final draft.

Yuhu Liang performed the experiments, analyzed the data, authored or reviewed drafts of the article, and approved the final draft.

Desmond Elliott performed the experiments, analyzed the data, authored or reviewed drafts of the article, and approved the final draft.

Anders Krogh conceived and designed the experiments, analyzed the data, authored or reviewed drafts of the article, and approved the final draft.

The following information was supplied regarding data availability:

The code is available at GitHub: https://github.com/ceegeeCode/randomodnar.

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
