# Peer review of "Context dependent prediction in DNA sequence using neural networks"

_PeerJ, doi:10.7717/peerj.13666_

## Round 0.1 · original submission · Major Revisions

Please address the reviewer's questions before the next round of submission.

Reviewer 1 ·

Basic reporting

Christian et al presented a neural network to predict DNA context which yields a better accuracy rate compared to the commonly used Markov model. The data were clearly structured and presented. This paper is clearly written, referenced well with a clear background introduction for the computation part, but I did not see any biological significance in the abstract and introduction part. I saw authors put their code on Git but can you provide some example running such as step-by-step Jupiter notebooks? This will allow for easy replication for readers.

Experimental design

None

Validity of the findings

None

Additional comments

None

Reviewer 2 ·

Basic reporting

The submitted manuscript tries to predict nucleotides from surrounding context using neural networks, and interpret the models using Fourier transform. Although this seems to be a detailed analysis, I am mostly concerned with the research question and methodology.

By reading the introduction (line 37-39), I still don’t know why prediction of nucleotides from their context is useful or meaningful. The motivation of this study is not clearly stated or explained. In addition, it seems that the models are tested on data on which they are trained, so I worry that the accuracy reported are inflated. The authors should make sure all performance metrics reported are unbiased estimates, meaning that the test set should not overlap with the training data.

The English language here is sometimes awkward and poses difficulty for understanding. Some examples include lines 73-75, 85-87, 121-122, 152-154, 162-163, 168-171, 175-176, 192-194, 321-323. There are also a few grammatical errors, such as lines 116, 119, 125, 194, 268-269. I suggest the authors use copy-editing services to polish your manuscript.

Experimental design

Below are some specific comments that may help the authors improve the manuscript.

Line 37-39: motivation is not clearly explained. Why is prediction of nucleotides from surrounding meaningful? What do you mean by structured parts of the genome?
Line 128-129: are you saying that the trained model is applied to all positions in a genome? If so, the performance is inflated: the model is trained on some part of the genome so testing on the same data will definitely leads to small errors. The model should be only evaluated on stretches of sequence that are not used for training at all. The only model that seems fairly evaluated is LSTM4S, where the test and train data are independent, but I am confused at the same time that you mention that “it was in the testing (also) applied to the even-numbered chromosomes”. When you say “also”, do you mean that you apply your trained model to both odd- and even-numbered chromosomes to get test performance? If so, LSTM4S was not fairly evaluated either.
Line 141-142: training set sounds very small as it is only 1/3 of genomes. Typically, you need most data for training so as to reduce variance of the models. If, for example, you use at least 2/3 for training, I think the predictions will be better.
Line 144: is flank size in bp? You may want to consider better naming of your models, as 1, 4, and 11 are not informative and readers will easily forget what they stand for. For example, since LSTM1 and LSTM4S use flank size of 200 and 50, I think renaming them to LSTM_200bp and LSTM_50bp is better.
Line 144-149: you mentioned that LSTM11 is identical to LSTM1 but trained less and the reason you include it was to investigate the effects of extra training. By trained less, do you mean LSTM11 is trained for less epochs? It comes as no surprise that undertrained models have lower performance, so I don’t think LSTM11 is worthy to be included as another model for comparison. In addition, you mention that LSTM4 has almost the same architecture as these models. Is it identical to LSTM1 or LSTM11? Is it fully trained or trained less? It is unclear from your description.
Line 153: since performance of models are overestimated, the actual performance of LSTM1 on unseen data should be less than 53%. That said, is LSTM1 really better than the simple/baseline models?
Figure 4: this is not a scatter plot but a density estimate.
Figure 5: please label x and y axes.
Line 204: please explain what the practical reasons are.
Line 201-236: the models only have ~50% accuracy so is the interpretation here reliable and relevant? If you want to study periodicity in DNA, why not studying the original sequences directly?
Line 213-214: can you show what Figure 5 looks like when you shuffle the positions? This way the readers can visualize the lack of periodicity on a random sequence.
Figure 6-7: please label x and y axes.
Line 280-282: you may want to briefly describe what you discover here.
Line 287-290: you can test this by varying the size of training set. For example, for the same genome, you can see what the test accuracy is when you use 1/10, 1/5, 1/3, … of the genome as training data, and use the same test set for evaluation. If the trend continues to increase, then this statement is probably true. If it plateaus early on, it means that small genome size is not the limiting factor here. In addition, it would nice if you can mention the genome size for each organism.

Validity of the findings

no comment

Reviewer 3 ·

Basic reporting

The manuscript in general is well-structured. Figures and supplementary materials are properly provided. Here are a few points that I believe it needs revision:

1. Title: “Prediction of DNA” sounds a little too generic. Maybe use terms like “(language) modeling of DNA sequences”, etc.

2. The language and writing need some overall improvement for better clarity, including but not limited to these examples:
Line 53~62: I feel that there are too many details provided in this section. Because these are not the results of this particular paper, I’d suggest writing about general conclusions of the cited work, instead of showing the actual numbers, e.g., “we observed that increasing context sizes improves our model performance…” and then specify the Markov model with k=14 which exceeds 50% accuracy.

Line 54~56: The explanation here seems to me a bit redundant and ambiguous. The “number of contexts of size 6” should refer to the number of all possible DNA sequences of length 6 (i.e. 4^6), which could be explicitly described here. Also, I’m not sure whether the “number of times each of the four letters occurs in the genome” is related to this number. Presumably, they should refer to the same number, but I cannot see from here.

Line 81~83: It’s great to mention some recent advances here, but there is no need to explain in detail the reason you haven’t tested DNABERT.

Line 120~123: The description of the model could be made more accurate and concise. You may refer to some neural network-related literature for how the output and architecture are described.

3. The idea of predicting missing parts in a sequence from the contexts is very similar to that of word2vec (arxiv.org/abs/1301.3781), which has actually been applied to DNA sequences, though at the basis of k-mers (arxiv.org/abs/1701.06279). It would be interesting to see how it could be compared with the authors’ methods in the introduction and/or discussion.

4. Line 191, 213, etc.: According to the writing guidelines, supplementary figures should be labeled and cited as “Fig. S1”, “Table S1”, etc. Also, as the supplementary figures are separated into several different files (especially the Fourier ones), I’d suggest merging some of them and using a hierarchical numbering (e.g. Fig. S1.1).

5. The axis labels and legends on Figures 4~8 are a bit too small to read. The subplots could be re-arranged (e.g., 2x2) so that texts can be a little larger. Also, maybe consider adding subplot numbering as well (e.g. Figure 8a) as it would make figure reference more straightforward.

6. Line 216: figure 4 is referred to here, but I don’t really think it is really related to this context.

7. Line 197: it says "disagreements are seen as bright areas off the diagonal". According to my understanding, the colors in figure 4 are related to densities of data points, so basically, all off-diagonal points are disagreements, while brighter areas are large patches of disagreements. The language here is a little misleading. Please clarify it a little bit.

8. For the first combined subplots in Figures 5~8, maybe try to use the same color scheme for the first subplot and the other three “separated” ones if possible.

9. Other minor issues
Line 269: should be LSTM.
The lines between 46 and 47 are not numbered.

Experimental design

The experimental design follows a clear logic flow, and I really appreciate the in-depth and comprehensive analysis of the learned patterns from the model, as well as the comparison between species. However, there are some details here that need to be either specified or carefully reviewed.

1. Train/test split
Please explain in greater detail how the train/test split is done, either the main text or the supplementary materials. Did you make sure that the training and test samples are from different non-overlapping segments? Otherwise, some test samples would be very similar to those in the training samples, which could potentially lead to information leaks. Please explain this in greater detail.

2. From the supplementary methods, I cannot see why the train/test split for LSTM1 is different, as all models seem to apply to the same set of training data. Moreover, it is not described elsewhere how the split for LSTM1 is different from the other models.

3. What is the reason that reverse complement is only considered for one of the models?

4. Naming of the models
Line 138~149: The numbering of the LSTM models is a bit confusing, and as they appear quite frequently throughout the paper, I’d suggest either (1) adding a table in this section showing their configurations or (2) simply numbering them according to the context size and training epochs (e.g., LSTM50, LSTM50-1). Also, “LSTM41” is mentioned in the caption of Table 1 but not elsewhere in the main text. It should be explained briefly here.

Validity of the findings

Most of the findings are properly supported by data and figures, but some need either further analysis or clarification to be made more convincing.
1. “Signals of periodicity in predictions” section (starting from line 200): the periodical model prediction is an interesting discovery. This may originate from frequently recurring sequence patterns. It would be beneficial to directly compare the level of GC contents, as well as enrichments of certain sequential motifs, between the contexts of the high-performing and low-performing positions.

2. Same section, figure 5: the description of the analysis and the caption of figure 5 seems inconsistent. The caption says “the following three plots show the segments covering the first half of the chromosome”. This leads to several confusions:
(1) Is there a reason why only the first half of the chromosome is shown? It is not mentioned anywhere else in the main text.
(2) I feel like the patterns in the left subplot (presumably showing all segments) actually look very similar to those on the right as if it is actually a simple combination of the latter (presumably showing half of the segments). Is it really the case as described in the caption?
(3) The result of “randomizing half of the positions”, as is stated in line 215, is not shown in figure 5, but the text refers to it. I assume it should not be the same thing as "segments covering the first half of the chromosome".

3. Same section, figure 7: between lines 250~253 the authors mention “two of the five peaks” and “the following three”, which are not very apparent by just looking at the figures. Maybe consider making a direct comparison plot with the peaks explicitly labeled.

4. Line 278, figure 8: the plot shows agreements between the prediction of the LSTM and the Markov model. With no ground truth included here, I don’t think it is sufficient to draw the conclusion that LSTM generally performs better. It only shows LSTM gives overall higher-confidence predictions. This conclusion can only be made by comparing the output of each model to the ground truth, then comparing their degree of agreement.

Reviewer 4 ·

Basic reporting

The current research focuses on investigating deep neural network models for DNA base prediction given neighbor context information. This is a parallel/follow-up work they carried out, in addition to the other project where they used conditional probability models (Markov models) for a similar prediction purpose. The topic being discussed, predicting DNA sequence using pure data-driven approaches, is relatively unique, and is therefore with limited previous literatures to provide background. To demonstrate their innovations, they, instead, used their own parallel work in Markov models as a baseline, which serves as a good context reference.

In their investigation, the authors tried different advanced Natural Language Processing (NLP) and Computer vision (CV) techniques and ended up using a series of hybrid models of Convolutional Neural Network (CNN) and bi-directional Long Short Term Memory units (LSTM). For other models that they could not try (e.g. Transformer-based models), either due to time limit or hardware resource limit, they provided a brief review and pointed to related references, which is appreciated and acceptable for such an exploratory project.

The manuscript is written in clear and professional English. Raw data, intermediate results, and original codes are provided where applicable. Figures are illustrative, but some may need adjustments to improve readability as detailed below. Overall the work is preliminary yet innovative and important for laying the foundation for introducing more advanced deep learning models for processing genomic data in the future.

Experimental design

The models, including the architectures, were developed mostly in a data-driven manner. A detailed description of model and experiment designs is included in the supplementary methods document. Codes, raw data recourses, and intermediate results are also provided for experiment replication.

The authors compared multiple variations of their CNN-LSTM models, as well as compared them with their baseline Markov model. Prediction and model performance comparisons were also made between chromosomes, annotations, and different organisms. Transfer learning was also considered (though not directly pointed out by the authors) by applying a mouse trained model on the human genome. Predictions’ patterns were further verified using Fourier transformation and the neural network model’s superiority over “simpler” models was established by doing a likelihood ratio test. In general, the experiment design is solid and systematic, though a few concerns remain as explained below, which may require further explanation or manuscript revision.

On line 128, as well as Supplementary Methods page 3, it is mentioned that the trained models were applied back to the full genome to make predictions, which included the training session. As well known, this may result in overfitting and misleading test performance metrics reported in the manuscript. The author explained that they used the LSTM4S model, which was trained only on the odd-number chromosomes and applied on the even-number chromosomes, to demonstrate there is no significant difference between the training and true test performances, and yet they also admitted on Line 162 that LSTM4S has an advantage on its training chromosomes. Therefore, this example does not explain well enough why it is unavoidable or should be using the whole genome for testing, except only the 2/3 not included in the training. Note that, even though it’s uncommon, it is not totally unacceptable to include the training data in testing, especially when dealing with genomic data. But to justify this usage, a further explanation about this training and test set-up is highly recommended.

Validity of the findings

The findings are most valid, and the Figures are in general clear and illustrative.

The authors compared their neural network models to their Markov models mostly, and sometimes referred to the random guess chance to demonstrate their models’ outperformance. However, the random guess baseline is not explicitly listed. It may be natural to think the random prediction accuracy would be 25% with four base types, but this is based on the assumption that A, C, G, and T are uniformly distributed across the genome, which may not be true, especially for some specific chromosomes. It is highly recommended that the authors could also report the distribution of the four bases in the datasets they used together with updated random prediction accuracy. Or the authors may instead use randomly shuffled data prediction performance as the baseline, which they mentioned they have done (Line 318) but the results are not included.

Some minor revisions recommended for Figures:
Figure 5, 6, and 7, the legends are a little bit hard to read. The authors may consider increasing the font size. In addition, axis labels and/or sub-plot titles are also needed for these figures and Figure 9.
Figure 8, Supplementary Figure 1, 3, and 4 may need to zoom in to make the labels readable. Supplementary Figure 2 also needs axis labels.

Additional comments

The manuscript is overall clearly written, the experiment design is solid and the results are comprehensive. The authors have been reasonably discreet about derivation and conclusions that they may draw from their results, also tried to find biological explanations for their findings whenever possible. This is highly appreciated for a computation-based biological and biomedical data analysis research as this one.

The authors have put much attention to the marginal improvement in accuracy of their neural network models in the Discussion section and seem to treat it as a limitation. Accuracy is actually just one aspect of the model performance. The overall confidence improvement in prediction probabilities (supported by Figure 4 and 8, and the likelihood-ratio test) is also very impressive. The authors may want to emphasize it in the Discussion section.

Currently, the models were trained and applied on the genome indistinguishably from the aspect of annotations. Since different types of regions have distinct statistical characteristics (as also pointed out in this research), a recommendation for future directions would be training different models for each specific annotation type. This is not required for this manuscript though.

---

## Round 0.2 · accepted · Accept

Congratulations! In the revised version, you addressed all concerns reviewers had. I think the new version is OK for publication now.

Reviewer 1 ·

Basic reporting

The authors have substantially improved the manuscript according to the reviewers' comments. I think this paper is suitable for publication now.

Experimental design

None

Validity of the findings

None

Reviewer 3 ·

Basic reporting

The reviewers have addressed all the confusions and concerns I pointed out.

Experimental design

The reviewers have addressed all the confusions and concerns I pointed out.

Validity of the findings

The reviewers have addressed all the confusions and concerns I pointed out.

Reviewer 4 ·

Basic reporting

The authors have properly addressed most of the points listed in my previous review and by other reviewers. Regarding the major concern that all reviewers shared, the one that there was no clear separation between the training and test data, the authors have supplemented figures and explanations to show that it has not resulted in biased model evaluation metrics. Though this may not be perfectly ideal, considering all the side-evidences that have been provided, the large data size and computation resource limitation, and the fact that having predictions available for the whole genome would benefit downstream genome-wide analyses and understanding, the way the authors address the issue is sufficient and acceptable. Therefore, I'd agree to "Accept" the manuscript.

Experimental design

NA

Validity of the findings

NA